# The Lebanese Red Algae *Jania rubens*: Promising Biomolecules against Colon Cancer Cells

**DOI:** 10.3390/molecules27196617

**Published:** 2022-10-05

**Authors:** Mariam Rifi, Zeina Radwan, Reem AlMonla, Ziad Fajloun, Jean Marc Sabatier, Achraf Kouzayha, Marwan El-Sabban, Hiba Mawlawi, Zeina Dassouki

**Affiliations:** 1Laboratory of Applied Biotechnology (LBA3B), AZM Center for Research in Biotechnology and its Applications, Doctoral School for Sciences and Technology, Lebanese University, Tripoli 1300, Lebanon; 2Department of Anatomy, Cell Biology, and Physiological Sciences, Faculty of Medicine, American University of Beirut, Beirut 1107 2020, Lebanon; 3Department of Biology, Faculty of Sciences 3, Lebanese University, Campus Michel Slayman Ras Maska, Tripoli 1352, Lebanon; 4CNRS UMR 7051, INP, Inst. Neurophysiopathol., Aix-Marseille Université, 13385 Marseille, France; 5Department of Biochemistry, Faculty of Sciences 3, Lebanese University, Campus Michel Slayman Ras Maska, Tripoli 1352, Lebanon; 6Faculty of Public Health III, Lebanese University, Tripoli 1310, Lebanon

**Keywords:** red algae, Jania rubens, biochemical analysis, antimicrobial, antioxidant, cytotoxic, human colon cancer cells

## Abstract

Colorectal cancer (CRC) is ranked the second most lethal type of tumor globally. Thus, developing novel anti-cancer therapeutics that are less aggressive and more potent is needed. Recently, natural bioactive molecules are gaining interest as complementary and supportive antineoplastic treatments due to their safety, effectiveness, and low cost. *Jania rubens* (*J. rubens*) is a red coral seaweed abundant in the Mediterranean and bears a significant pharmacological essence. Despite its therapeutic potential, the natural biomolecules extracted from this alga are poorly identified. In this study, the proximal analysis revealed high levels of total ash content (66%), 11.3% proteins, 14.5% carbohydrates, and only 4.5% lipids. The elemental identification showed magnesium and calcium were high among its macro minerals, (24 ± 0.5 mg/g) and (33 ± 0.5 mg/g), respectively. The Chlorophyll of *J. rubens* was dominated by other pigments with (0.82 ± 0.02 mg/g). A 2,2-diphenyl-1-picrylhydrazyl (DPPH) radical scavenging assay identified effective antioxidant activity in various *J. rubens* extracts. More importantly, a 3-(4,5-dimethylthiazol-2-yl)-2,5-diphenyltetrazolium bromide (MTT) tetrazolium reduction and wound healing assays indicate that organic extracts from *J. rubens* significantly counteract the proliferation of colon cancer cell lines (HCT-116 and HT-29) and inhibit their migratory and metastatic properties in a dose and time-dependent manner. Overall, this study provides insight into the physicochemical properties of red seaweed, *J. rubens*, and identifies its significant antioxidant, cytotoxic, and anti-migratory potential on two colorectal cell lines, HCT-116 and HT-29.

## 1. Introduction

Algae are known worldwide as an exciting field of research and an essential source of active constituents with various applications, including in the pharmaceutical, cosmetic, food, and, agricultural industries [1]. Seaweeds are rich in nutrients such as amino acids, fatty acids, lipids, vitamins, and essential minerals [2,3]. In addition, they contain important biogenic metabolites such as polysaccharides, phenols, flavonoids, and carotenoids [4,5,6]. More importantly, accumulating evidence suggests that compounds extracted from seaweeds induce antitumor effects through multiple mechanisms of action, including the inhibition of cancer cell growth, metastasis, and the induction of apoptosis [7].

The Lebanese coast is affluent in seaweed, with more than ninety-four macroalgae belonging to the different algal groups (green, brown, and red algae) [8]. The biodiversity of the Lebanese coast is related to its location at the warmest corner of the Mediterranean Sea and its proximity to the northern gate of the Suez Canal, which allows the interaction between the Red and the Mediterranean Sea [6,9]. Rhodophyta or red algae are the sources of more than 1658 natural products [10]; *J. rubens* is the most common red seaweed in the eastern part of the Mediterranean. Different studies have been performed around the Mediterranean coast to identify the bioactive components of *J. rubens* and their biological effects. A survey completed in Egypt showed that *J. rubens* of the Alexandrian coast has a molluscicidal impact [2]. Extracts of *J. rubens* collected from the Izmir coast in Turkey and from the Tunisian coast near Bizerte have antimicrobial effects [11,12]. More importantly, studies on *J. rubens* of the Mediterranean Sea in Alexandria, Egypt, and the Red Sea in Hurghada, Egypt have identified different bioactive components with cytotoxic effects [13,14,15]. However, *J. rubens* of the Lebanese coast has not been investigated yet.

According to the National Comprehensive Cancer Network’s (NCCN) latest review, CRC is the second leading cause of cancer mortality and the fourth most identified cancer in the United States of America (USA). In the MENA region (the Middle East and North Africa region), Lebanon has the highest cancer incidence [16]. The standard CRC treatments are surgery, chemotherapy, and radiation or a combination of more than one mode depending on the stage, location, metastasis, and the patient’s health condition [17]. Traditional chemotherapeutic agents aim to inhibit tumor progression and metastasis; however, they also affect the tumor cell environment and lead to serious side effects [18]. Targeted therapy has increased the survival rate in CRC patients; however, it may have limited efficacy in some patients and may develop drug resistance [19]. To reduce the complications of conventional cancer treatments, researchers have headed towards more natural compounds with lower costs and fewer side effects that can increase the potential for complete recovery [18,20]. Specifically, interest in seaweeds has increased as reliable alternative compounds that can assist cancer treatment [21]. However, the research on seaweed-derived bioactive agents and their antitumor potentials is still in its infancy [22]. Although *J. rubens* is highly abundant on the Lebanese coast, this alga has been poorly studied. In this study, we will identify the physicochemical properties of *J. rubens* extracted from the northern Lebanese coast. We will also research the antioxidative, bactericidal, cytotoxic, and anti-migratory potential of different *J. rubens* extracts against colon cancer.

## 2. Results

### 2.1. Physicochemical Analysis of J. rubens

The physicochemical evaluation of *J. rubens* will allow its standardization as raw material, and it will help quantify many elements in this seaweed, which will identify its nutritional value. The pie chart (Figure 1) reflects the variation in the organic content and inorganic ash present in *J. rubens* on the Lebanese coast. The proximal analysis of *J. rubens* powder shows that proteins (11.3%) and carbohydrates (14.5%) were relatively higher than lipids (4.5%).

In addition, the inorganic ash of this algae was found to be very abundant (66%), and it is mainly composed of equal amounts of water-soluble (23.26%) and acid-insoluble ash (24%) (Table 1).

The elemental analysis, presented in (Table 2), showed the elevated levels of some essential macro-minerals in this seaweed: calcium (33mg/g) and magnesium (24 mg/g). Furthermore, the photosynthetic pigment, chlorophyll (1.04 mg/g), dominated the other pigments in *J. rubens* (Table 3).

### 2.2. The Phenol and Flavonoid Content of J. rubens Extracts

The concentration of the phenols and flavonoids in *J. rubens* varied from one extract to another. The highest phenolic content was recorded in methanol (M) Soxhlet, with (24.3 ± 0.25 mg/g). However, the flavonoid content was highly abundant in the dichloromethane/methanol (DM, 1:1) crude organic extract (42.29 ± 0.6 mg/g), followed by the M Soxhlet extract (28.8 ± 2.5 mg/g) (Figure 2).

### 2.3. The Antioxidant Effect of J. rubens Extracts

The antioxidant properties of the aqueous and organic extracts of *J. rubens* were investigated by DPPH free radical scavenging activity. The standard used was vitamin C. The results showed that 750 µg.mL^−1^ of the DM Soxhlet extract has the highest antioxidant activity (64.62 ± 6.68%) (Figure 3A). At a concentration of 750 µg.mL^−1^, the scavenging activity of the M crude extract and the AQ Soxhlet extract was 61.52 ± 5.44% and 60.44 ± 0.54%, respectively (Figure 3B and Figure 3C). The extraction methods had no significant influence on the antioxidant activity.

### 2.4. Antibacterial Effect of J. rubens Extracts

The antibacterial activity of the DM, M, and AQ extracts was evaluated against some Gram-positive and Gram-negative bacteria using the agar disc diffusion method (Table 4). Penicillin/streptomycin was used as a positive control. The results showed that *J. rubens* extracts have no antibacterial activity against any of the bacteria species tested.

### 2.5. Anti-Proliferative Activity

#### 2.5.1. *J. rubens* Extracts Inhibit the Proliferation of HCT-116 and HT-29 Cells

Effect of DM extracts on the viability of CRC cell lines

In the present study, an MTT assay was used to evaluate the impact of *J. rubens* DM extracts’ (0–750 µM at 24 h and 48 h) treatment on the cell viability of two CRC cell lines, the HCT-116 and HT-29 colon cancer cells (Figure 4). The results showed that DM extracts significantly decreased the cell viability (*p* < 0.0001) of both treated cell lines at all tested concentrations. Interestingly, the results indicate that DM Soxhlet exhibited a much higher cytotoxic effect compared to the DM crude extract. The decrease in HCT-116 cell viability treated with DM Soxhlet compared with the non-treated control cells at concentrations of 100–250–500–750 µg.mL^−1^ reached 68%, 65%, 56%, and 32%, respectively, while it was 88%, 87%, 82%, and 78% upon the treatment with DM crude, at the same time point (24 h). Besides, the DM Soxhlet and crude extracts exhibit a time-dependent inhibitory effect on HCT-116 cells. Upon treatment of the cells with 750 µg.mL^−1^ of DM Soxhlet, the percentage of cell proliferation was 32.11 ± 5.71% at 24 h, to reach only 4.93 ± 1.85% at 48 h. Similarly, DM crude (750 µg.mL^−1^) decreased the cell viability from 78.31 ± 4.45% to 31.99 ± 6.88% at 24 and 48 h, respectively. Further analysis of the results revealed that the DM Soxhlet treatment exhibited a dose-dependent effect on the HCT-116 cells at both time points, 24 and 48 h of treatment (0.05 < *p* < 0.0001), while DM crude induces a dose-dependent inhibition only after 48 h of treatment (*p* < 0.0001). Overall, the results indicate that DM Soxhlet is more potent than the crude extract. In addition, the HCT-116 cells (IC_50_ equal 499.94 µg.mL^−1^) were more sensible than the HT-29 (531.44 µg.mL^−1^) to treatment (Table 5). The results were validated with the trypan blue assay (Appendix A).

Effect of M extracts on the viability of CRC cell lines

Since M extract is more polar than DM extract and different molecules could be extracted with solvents of a different polarity [23], using the same experimental conditions, our next objective was to investigate the antiproliferative property of the M Soxhlet and crude extracts of *J. rubens* on the two treated colon cancer cells. The results showed that this extract, both Soxhlet and crude, significantly decreased the cell viability (*p* < 0.0001 compared to control) at all tested concentrations (Figure 5). Moreover, M Soxhlet and crude revealed a time-dependent inhibition (0.01 < *p* < 0.0001). Upon treatment with 750 µg.mL^−1^ of the M Soxhlet extract, the inhibition of the cell viability was 57.39 ± 5.41% at 24 h and further reduced by 47% to 9.54 ± 4.09% at 48 h. Remarkably, M Soxhlet is more potent on HCT-116 cells than HT-29 with IC_50_ equal to 389.73 µg.mL^−1^ and 666.82 µg.mL^−1^, respectively.

Effect of AQ Extracts on the Viability of CRC Cell Lines

We also tested the cytotoxic effect of the aqueous extract. The AQ extracts induced a significant antiproliferative effect, with *p* < 0.0001 on both CRC cell lines (Figure 6). However, the results did not reveal any major difference between the two tested cell lines, and the cytotoxic effect of the AQ extracts is neither dose- nor time-dependent. Overall, the cytotoxicity of the AQ extracts is not as pronounced as the one observed with DM and M.

Subsequently, considering all the results up to this point, we concluded that organic extracts are generally more potent than aqueous extracts on both cell lines, and the Soxhlet extraction method increases the cytotoxic potential of the extracts. Also, the HCT cell line is more sensitive than HT-29 to the organic extract’s treatment.

#### 2.5.2. DM Soxhlet and M Soxhlet Extracts of *J. rubens* Reduce the Migration Ability of HT-29 and HCT-116 Cells

The Effect of DM Soxhlet extracts on the migration ability of HT-29 and HCT-116 cells

Since DM and M Soxhlet extracts presented a significant decrease in cell proliferation by MTT assay, we investigated the anti-migratory potential of these extracts on HCT-116 and HT-29 cells by a classic wound-healing assay.

The DM Soxhlet treatment showed remarkable and significant inhibition of cell migration in a dose-dependent manner (0.01 < *p* < 0.0001). At 24 h, the HCT-116 cells were treated with 100 µg.mL^−1^ of DM Soxhlet, which migrated at a rate of 22.4% compared to 37.5% for the control; this ratio decreased to 6.11% with 750 µg.mL^−1^ of the DM Soxhlet treatment (Figure 7). Similarly, a concentration of 750 µg.mL^−1^ significantly decreased the gap closure rate of the HT-29 cells to 9.59% at 24 h, a percentage four times lower than the control. Thus, DM Soxhlet exerts a potent anti-migratory effect on both cell lines after 24 h.

Anti-migratory effect of M Soxhlet extracts against HCT-116 and HT-29

Regarding the M Soxhlet treatment, the quantitative analysis of the wound area showed that the cells treated with different concentrations (100–750 µg.mL^−1^) exhibited a significant dose-dependent anti-migration effect at 24 h (0.01 < *p* < 0.0001). The percentage of wound-healing decreased to 18.11% and 7.89% at 100 µg.mL^−1^ and 750 µg.mL^−1,^, respectively, compared to 36.28% for the control in the HCT-116 cells (Figure 8). We noted that M Soxhlet showed a significant anti-migration effect at an early point (6 h) at 750 µg.mL^−1^ (9.27% for HCT-116 and 8.85% for HT-29) compared to 25% for the control.

## 3. Discussion

In this study, we first identified the physiochemical characteristics and the antioxidant, cytotoxic, and anti-migratory effects of Lebanese red seaweed, *J. rubens*. We started by measuring the ash content. Our study revealed that the total ash content of Lebanese *J. rubens* is 66 %. This amount is considered high compared to other red seaweeds [24] and indicates abundant minerals and nutraceutical products [25,26]. Then, we analyzed the organic content, and we showed that Lebanese *J. rubens* contain 11.3% proteins and 14.48% carbohydrates. The protein and carbohydrate active biomolecules could have a role in the anticancer and antioxidant effects recorded in this study. Indeed, previous studies have shown that algal proteins possess significant biological potential as anti-inflammatory, antibacterial, and antioxidant compounds [27,28]. Moreover, the proteins extracted from red algae also have a potential antitumor effect [29,30]. Phycocyanin, a phycobiliprotein present in red algae, possesses an effective anticancer effect against MCF-7 breast cancer cells [31], HT29 colorectal carcinoma cells [32], A549 lung adenocarcinoma cells [33] and others. Moreover, many reports have indicated that red seaweed polysaccharides exert an anticancer effect by activating the immune system and increasing the infiltration of the immune cells into the tumor [34,35]. For example, *Porphyra haitanensis* polysaccharides exerted inhibitory effects on growth in the HT-29, LoVo, and SW-480 colon cancer cell lines [36].

Moreover, the present study revealed that Lebanese *J. rubens* is rich in calcium (24 ± 0.5mg/g) and magnesium (33 ± 0.5 mg/g). These elements are essential for numerous human metabolic reactions, such as enzymatic regulation and the metabolism of lipids, carbohydrates, and proteins [37,38]. Also, the high amount of calcium and magnesium in marine seaweeds plays a role in suppressing the growth and differentiation of colon cancer carcinoma [39]. Calcium is a second messenger in many pathways related to cell proliferation and death [40]. Previous studies showed that many natural components induce anticancer properties through Ca-mediated pathways [41,42,43]. Also, magnesium plays a vital role in many physiological activities and promotes an anti-tumor activity by inducing apoptosis, autophagy, and cell cycle arrest [44]. Similarly, clinical studies revealed that magnesium protects against colorectal cancer by inhibiting the expression of c-myc and the ornithine decarboxylase activity in the intestine’s mucosal epithelium [45].

We have also shown that *J. rubens* collected from the Lebanese coast has a four times higher chlorophyll content than the same algae collected from the Egyptian coast (0.25 mg/g) [46]. In general, the chlorophyll content in seaweeds depends on the depth of the algae. The deeper the location, the higher the chlorophyll amount [47]. In our study, *J. rubens* was collected at 2–3 m deep, and this could explain the high amount of chlorophyll. The role of seaweed pigments is not only to characterize each group of algae but also it has an important role as an antioxidant, anti-inflammatory, and anticancer [47,48,49,50]. Previous studies showed that chlorophyll and carotenoids inhibit cancer proliferation and can induce apoptosis in different cell lines, including colon cancer cells (HT-29, Caco-2), breast cancer cells (MCF-7), T-cell leukemia, and others [51,52,53].

Phytochemicals, such as flavonoids and phenols, are produced by seaweeds [54]. Phenolic compounds are found in large amounts in red seaweeds compared to other groups of macroalgae [55,56]. Phenols are known for their importance as hormones, antioxidants, cofactors, and anti-tumor compounds [3,57]. Our present data show that all organic extracts are rich in flavonoids and phenols, while aqueous extracts contain low levels. The difference in the flavonoid and phenols content between these extracts is due to the polarity of the solvent, the method of extraction, the molecular weight of the phenolic components, the temperature, and the extraction [58]. Some reports demonstrated that flavonoids and phenols work as antioxidants; they can scavenge reactive oxygen species and inhibit lipid peroxidation [59,60]. Furthermore, phenols have been marked as a substantial metabolite for anti-tumor activity [61]. Some studies showed that polyphenols promote anticancer activity via different pathways, including apoptosis, cell cycle arrest, and metastasis [62]. In addition, phenolic compounds possess an anti-migratory effect against H460 lung cancer cells [63]. Our data revealed that phenols content is the highest in M Soxhlet extract (24.26 ± 0.25 mg/g), which could explain its antioxidant, antiproliferative, and anti-migratory potentials against colorectal cancer cells.

In addition, the highest antioxidant activity detected, among all extracts, was in the DM Soxhlet one with 64.6% (±6.7). In fact, there is a correlation between its antioxidant and anticancer effects [61,64,65]. Antioxidants inhibit carcinogenic agents, modulate cancer cell signaling, and induce apoptosis and cell cycle arrest [66]. Many studies have shown that red seaweeds exhibit high antioxidant activity, leading to an antiproliferative effect and apoptosis against different cancer cells [67,68]. We believe that the high antioxidant effect of DM Soxhlet plays a role in its effective cytotoxic and anti-migratory properties.

Previous studies showed that seaweeds contain bioactive components with antimicrobial effects. Lebanese *J. rubens* has no antibacterial effect against bacteria. However, *J. rubens*, collected from Tunisia and Egypt, showed high potential in inhibiting the growth of different Gram+ and Gram- bacteria [11,69]. This variation may be due to environmental factors since seaweeds are known to change their active components in response to changes related to season, temperature, pollution, and others [1].

Novel CRC treatments are investigated to reduce the side effects and prolong patient survival. The new treatment methodology must include cytotoxicity to limit cancer progression and development, the anti-migratory effect to limit secondary metastasis, and the antioxidant potential to limit cancer-related oxidative stress. The cytotoxicity and antimetastatic effect of Lebanese *J. rubens* extracts are associated with the presence of anticancer metabolites, including the phenols, flavonoids, and antioxidants detected in this study. Interestingly, DM and M Soxhlet inhibit the cell proliferation of CRC cells more than the crude extracts. Thus, extraction at high temperatures by the Soxhlet apparatus accumulates more bioactive molecules with anti-tumor activity [6]. This concept may be attributed to many reasons, such as the molecules’ polarity, size, and lipophilicity [70,71]. In addition, organic extracts (DM and M) are more potent than aqueous extracts. Therefore, we believe that the cytotoxic molecules are more soluble in organic extracts. This result emphasizes the influence of the choice of solvent extraction on the cytotoxic activity of seaweed extracts against different cancer cell lines [23].

For the first time, we studied the effect of Lebanese *J. rubens* extracts on colon cancer cell migration. Interestingly, DM and M Soxhlet extracts inhibit the migration of cancer cells after 24 h of treatment. Considering all previous results, we believe that DM and M Soxhlet extracts can be used as potential adjuvant treatments in addition to conventional chemotherapy.

## 4. Materials and Methods

### 4.1. Macroalgal Biomass

*J. rubens* samples were collected from the North Lebanese coast of the Mediterranean region at a depth of 2–3 m. Fresh seaweeds were rinsed thoroughly and air-dried at room temperature, then ground to a fine powder. The herbarium voucher of *J. rubens* (AZM-1105) was preserved at the Doctoral School of Science and Technology, Lebanese University.

### 4.2. Physicochemical Analysis

#### 4.2.1. Proximal and Elemental Analysis

Lipids, moisture, total ash, acid-insoluble, and water-soluble ash were quantified according to the methods described by the Association of Official Analytical Chemists [72]. Ashed seaweeds were moistened with distilled water and then dissolved with nitric acid and deionized water for macro and micro elemental analysis. The solution was heated at 100 °C till dry; then, the residue was ashed at 500 °C for one h. Later, the ash was dissolved with HCl and filtered with ashless filter paper. Atomic absorption spectrometry was used, and for each element measured, a standard calibration curve was prepared.

#### 4.2.2. Organic Content Analysis

For total carbohydrates analysis, algal samples were hydrolyzed with concentrated sulfuric acid at 37 °C for one hour, and the acid strength was diluted to 1 M, followed by two hours of boiling. The Dubois phenol–sulfuric acid method was used, and glucose was used as a standard [73]. The absorbances were read at 490 nm.

For lipid analysis, the dried seaweeds were extracted with chloroform/hexane (2:1) and stirred overnight. The mixture was centrifuged, and the collected supernatant containing lipids was evaporated by rotavap, whereas the residue was saved for protein extraction. The lipid yield was determined gravimetrically.

For proteins, ultra-pure water (40 °C) was added to the residue obtained from the previous lipid extraction. The solution was stirred for 24 h at 40 °C, filtered, and then hot ultra-pure water was added. Proteins were precipitated by zinc sulfate and barium hydroxide, followed by centrifugation at 5000× *g* for 10 min at 4 °C. The resulting protein pellet was lyophilized and quantified using the Bradford method, with bovine serum albumin (BSA) as a standard.

#### 4.2.3. Pigment Analysis

Pigment extracts were isolated using 80% acetone extraction. The extracts were utilized for chlorophyll (Chl) estimation. Using Arnon’s equations, the absorbance was read at 645 and 663 nm in an ultraviolet spectrophotometer [74]. Carotenoids were estimated by the method of Kirk and Allen (Kirk and Allen 1965) [75], where the same extract was measured at 480 nm in the spectrophotometer.

### 4.3. Seaweed Solvent Extraction

For crude and Soxhlet extractions, three different solvents were used. These include dichloromethane/methanol (DM, 1:1), methanol (M), and finally, water or aqueous (AQ) solvents. For crude extraction, powdered algae were macerated with each solvent for three days at room temperature in an orbital shaker. Powdered algae were also extracted with the same solvents at an elevated temperature using a Soxhlet extractor for six hours. All extracts were concentrated using a rotary evaporator, and the aqueous extracts were lyophilized.

### 4.4. Total Phenol Content

The extract’s total phenolic content (TPC) was determined by the Folin–Ciocalteu method [76]. The extract (100 μL) aliquot was mixed with 750 μL of a ten-fold diluted Folin–Ciocalteu’s phenol reagent. After 5 mins, a 7.5% sodium bicarbonate solution was added and then the reaction was allowed to stand for 90 min at room temperatures [77]. The absorbance was measured at 725 nm with a gallic acid standard curve for estimating the TPC concentration in the sample. The phenolic content was calculated as mg gallic acid equivalents GAE per gram of dry algal powder.

### 4.5. Total Flavonoid Content

The aluminum chloride colorimetric assay was implemented to determine the total flavonoid content of algal samples [78]. Each extract’s aliquot (0.5 mL) was mixed with 1.5 mL of ethanol and 0.1 mL of 10% aluminum chloride (10%). Then, 0.1 mL of potassium acetate (1M) was added, followed by 2.8 mL of distilled water. The mixture was incubated for 30 mins at room temperature. The absorbance was measured at 415 nm, where quercetin was used as a standard. The concentration of total flavonoid content was expressed as mg quercetin equivalent (QE) per gram of dried algal material.

### 4.6. DPPH Free Radical Scavenging Assay

Seaweed extracts were aliquoted into concentrations (100–750 μg/mL). A methanolic DPPH solution was added to samples and incubated in the dark for 30 min. The absorbance was measured at 517 nm [77]. Vitamin C was used as a standard, and the percentage of free radical scavenging was calculated using the formula:
Free radical scavenging (%) = ([control OD − sample OD]/control OD])/100

### 4.7. Antimicrobial Assay

Antibacterial tests were performed using the agar disc diffusion method. Sterile discs were impregnated with *J. rubens* extracts of a concentration of 1 mg/ml deposited on the surface of the agar medium (Mueller–Hinton Agar, pH 7.4 ± 0.2 at 25 °C) that was previously inoculated with various bacteria strains. The five bacterial strains were obtained from the Health and Environment Microbiology Laboratory of AZM Center, Lebanon: *Escherichia coli*, *Pseudomonas aeruginosa, Streptococcus pneumonia, Bacillus cereus,* and *Acinetobacter baumannii*. The results were expressed by measuring the diameters of the bacterial inhibition zone.

### 4.8. Cell Lines and Culture

Colon cancer cells (HT-29 and HCT-116) were purchased from the American Type Culture Collection (ATCC). Cells were cultured in DMEM in a humidified incubator at 37 °C, with 5% CO_2_ and 95% air. The media was supplemented with 10% heat-inactivated fetal bovine serum and 1% penicillin/streptomycin (100 U·mL^−1^).

#### 4.8.1. Cell Viability Assay

The cell viability assay is an MTT-based method that measures the ability of metabolically active cells to convert tetrazolium salt into formazan blue. HCT-116 and HT-29 cells were seeded overnight in a 96-well plate, to be later treated with different concentrations of *J. rubens* extracts (100–750 μg.mL^−1^). Cells were treated at two-time points (24 and 48 h), then incubated with MTT for two hours at 37 °C in the dark. The ELISA microplate reader measured absorbances at 570 nm. All determinations were carried out in triplicate.

#### 4.8.2. Trypan Blue Test

Colon cancer cells (HCT-116and HT-29) were seeded at a density of 5 × 104 in a 24-well plate. Based on MTT data, only potent organic extracts were tested to confirm their cytotoxicity using a trypan blue assay: DM and M Soxhlet extracts. After 24 and 48 h, treated and non-treated cells were washed, trypsinized, and stained with trypan blue (0.4%). A hemocytometer, using a light microscope, counted the number of viable versus dead cells. All determinations were carried out in triplicate.

#### 4.8.3. Wound-Healing Migration Assay

HCT-116 and HT-29 cells were seeded in a 24-well plate overnight. First, a scratch wound was applied with a sterile 200 μL tip at confluency. Cell debris was washed twice; then, cells were incubated with the potent organic extracts at different concentrations (0, 100, 250, 500, and 750 µg.mL^−1^). Images of the wounds were taken post 0, 6, and 24 h treatments. Images were captured using a digital camera coupled to a light microscope. The Image J analysis program analyzed the surface area.

### 4.9. Reagents

We used, in this study, the following reagents: Sulfuric acid (Sigma-Aldrich, St. Louis, MO, USA), nitric acid (Sigma Aldrich), hydrochloric acid, glucose, zinc sulfate, barium hydroxide, potassium acetate (Laboratory chemicals-Lebanon), bovine serum albumin (Sigma-Aldrich CAS 9048-46-8), acetone (Supelco), dichloromethane (Sigma-Aldrich), methanol (Sigma-Aldrich), Folin–Ciocalteu phenol (Sigma-Aldrich CAS 47641), ethanol (Sigma -Aldrich), aluminum chloride (Sigma-Aldrich), 2,2-diphenyl-1-picrylhydrazyl (Sigma-Aldrich), quercetin (Sigma-Aldrich), Dulbecco’s Modified Eagle’s Medium (Sigma-Aldrich CAS D5796), fetal bovine serum (Sigma-Aldrich CAS F9665), phosphate-buffered saline (Sigma-Aldrich, CAS 806552), penicillin/streptomycin (Sigma Aldrich, P4333), 3-(4,5-dimethylthiazol-2-yl)-2,5-diphenyltetrazolium bromide (Sigma-Aldrich CAS M56655), and trypsin (Sigma Aldrich CAS T3924).

### 4.10. Statistical Analysis

All statistical analyses (*t*-test, one-way ANOVA, and two-way ANOVA) were performed using GraphPad Prism 7 (version 7.0, USA). Probability values below 0.05 (* *p* < 0.05) were considered as significant, and values below 0.01 (** *p* < 0.01), 0.001 (*** *p* < 0.001), and 0.0001 (**** *p* < 0.0001) were considered as highly significant. The quantitative data are expressed as means ± SD from the indicated set of experiments.

## Figures and Tables

**Figure 1 molecules-27-06617-f001:**
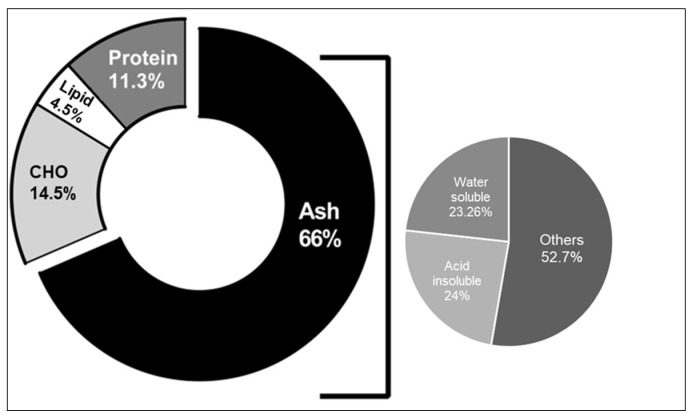
Pie chart showing the organic (proteins, lipids, and carbohydrates) and ash (water-soluble and acid-insoluble) content of the Lebanese red seaweed, *J. rubens*.

**Figure 2 molecules-27-06617-f002:**
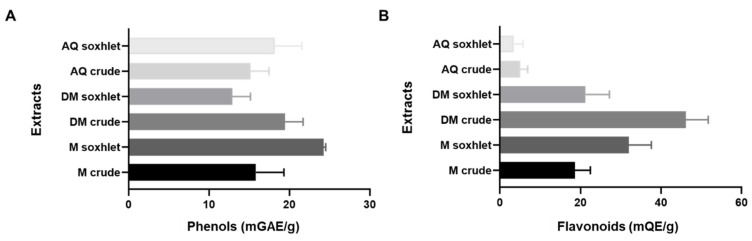
Total phenolic and flavonoid content of *J. rubens* various extracts. (**A**) The extract’s total phenolic content was determined by the Folin–Ciocalteu method and calculated as mg gallic acid equivalents GAE per gram of dry algal. (**B**) The extract’s total flavonoid content was measured by an aluminum chloride colorimetric assay and expressed as mg quercetin equivalent (QE) per gram of dried algal material. Values are reported as the mean ± SD (*n* = 3). AQ: aqueous; DM; dichloromethane/methanol; M: methanol.

**Figure 3 molecules-27-06617-f003:**
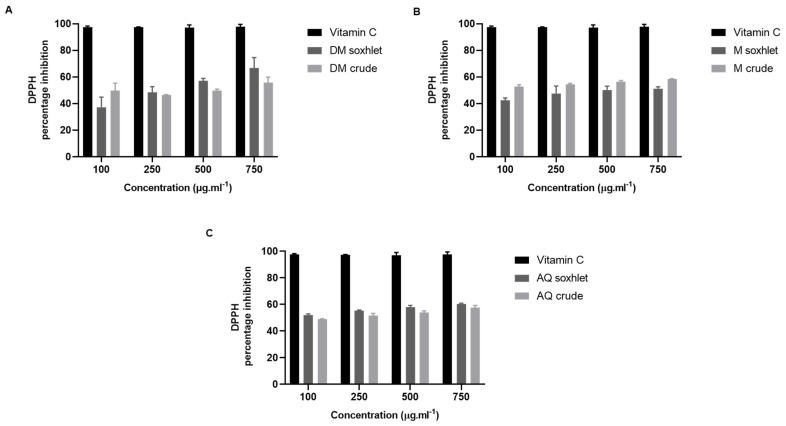
The antioxidant activity of *J. rubens* extracts: Determination of DPPH scavenging activity of different concentrations of *J. rubens* extracts (100, 250, 500, and 750 µg.mL^−1^). DPPH percentage inhibition of (**A**) DM extracts: DM Soxhlet and DM crude; (**B**) M extracts: M Soxhlet and M crude; (**C**) AQ extracts: AQ Soxhlet and AQ crude. Vitamin C was used as the standard. The absorbance was measured at 517 nm. Values are reported as the mean ± SD (*n* = 3). DM: dichloromethane/methanol (1:1); M: methanol; AQ: aqueous.

**Figure 4 molecules-27-06617-f004:**
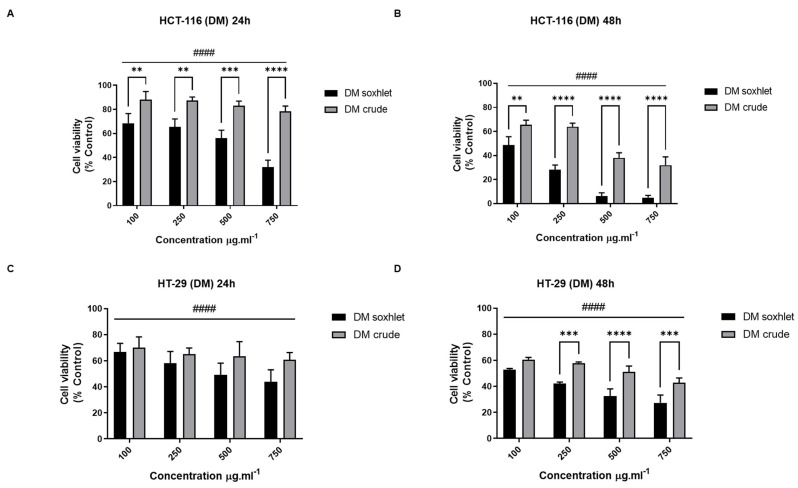
Effect of DM extracts on the viability of CRC cell lines. HCT-116 and HT-29 cells were seeded in 96-well plates and treated subsequently with different concentrations of DM extracts (0–100–250–500–750 µg.mL^−1^). MTT tetrazolium reduction assay was used to measure HCT-116 and HT-29 cell viability after 24 h and 48 h treatment. (**A**,**B**) Effect of DM extract (Soxhlet and crude) on HCT-116 cells at 24 h and 48 h. (**C**,**D**) Effect of DM extract (Soxhlet and crude) on HT-29 at 24 h and 48 h. The percentage of cell viability was calculated considering the value of the control as 100%. Results are presented as the mean ± SD (*n* ≥ 3). ** *p* < 0.01, *** *p* < 0.001, and **** *p* < 0.0001 vs. crude extract. ^####^
*p* < 0.0001 vs. control group. DM: dichloromethane/methanol.

**Figure 5 molecules-27-06617-f005:**
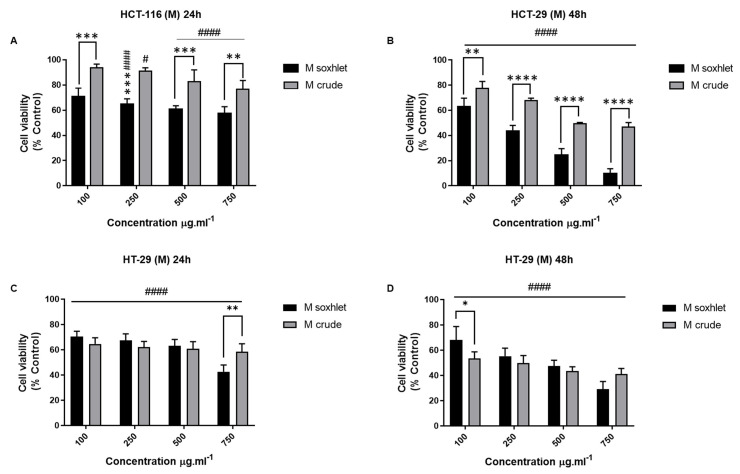
Effect of M extracts on the viability of CRC cell lines. HCT-116 and HT-29 cells were seeded in 96-well plates and treated subsequently with different concentrations of M extracts (0-100-250-500-750 µg.mL^−1^). MTT tetrazolium reduction assay was used to measure HCT-116 and HT-29 cell viability after 24 h and 48 h treatment. (**A**,**B**) Effect of M extract (Soxhlet and crude) on HCT-116 cells at 24 h and 48 h. (**C**,**D**) Effect of M extract (Soxhlet and crude) on HT-29 at 24 h and 48 h. The percentage of cell viability was calculated considering the value of the control as 100%. Results are presented as the mean ± SD (*n* ≥ 3). * *p* < 0.05, ** *p* < 0.01, *** *p* < 0.001, and **** *p* < 0.0001 vs. crude extract. ^####^
*p* < 0.0001 vs. control group. M: methanol.

**Figure 6 molecules-27-06617-f006:**
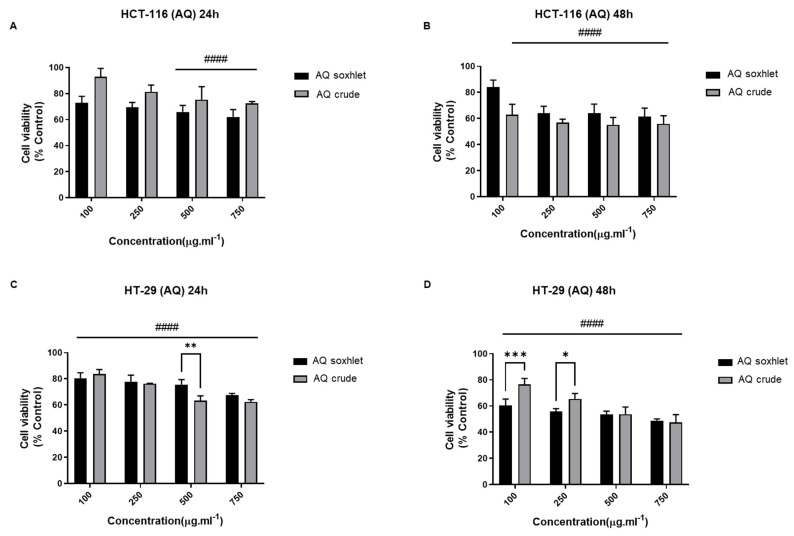
Effect of AQ extracts on the viability of CRC cell lines. HCT-116 and HT-29 cells were seeded in 96-well plates and treated subsequently with different concentrations of AQ extracts (0-100-250-500-750 µg.mL^−1^). MTT tetrazolium reduction assay was used to measure HCT-116 and HT-29 cell viability after 24 h and 48 h treatment. (**A**,**B**) Effect of AQ extract (Soxhlet and crude) on HCT-116 cells at 24 h and 48 h. (**C**,**D**) Effect of AQ extract (Soxhlet and crude) on HT-29 at 24 h and 48 h. The percentage of cell viability was calculated considering the value of the control as 100%. Results are presented as the mean ± SD (*n* ≥ 3). * *p* < 0.05, ** *p* < 0.01, *** *p* < 0.001, and **** *p* < 0.0001 vs. crude extract. ^####^
*p* < 0.0001 vs. control group. AQ: aqueous.

**Figure 7 molecules-27-06617-f007:**
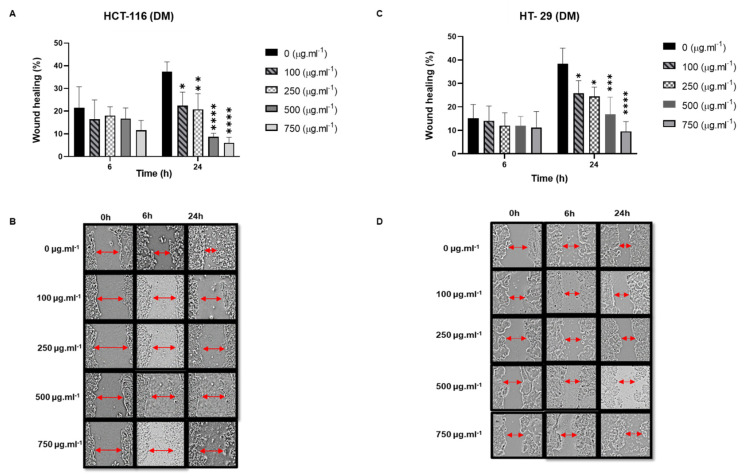
Wound healing assay showing that DM Soxhlet (100–250–500–750 µg.mL^−1^) inhibits the migration of HCT-116 and HT-29 cells after treatment for 24 h. (**A**,**B**) DM Soxhlet treatment of HCT-116: (**A**) Percentage of wound-healing at 6 h and 24 h. (**B**) Images of scratched HCT-116 cells taken at 0 h, 6 h, and 24 h. (**C**,**D**) DM Soxhlet treatment of HT-29. (**C**) Percentage of wound-healing at 6 h and 24 h. (**D**) Images of scratched HT-29 cells taken at 0 h, 6 h, and 24 h. Representative images were captured by a Leica microscope. Data are reported as mean ± SD, *n* = 3. * *p* < 0.05, ** *p* < 0.01, *** *p* < 0.001, and **** *p* < 0.0001 vs. control group. DM: dichloromethane/methanol.

**Figure 8 molecules-27-06617-f008:**
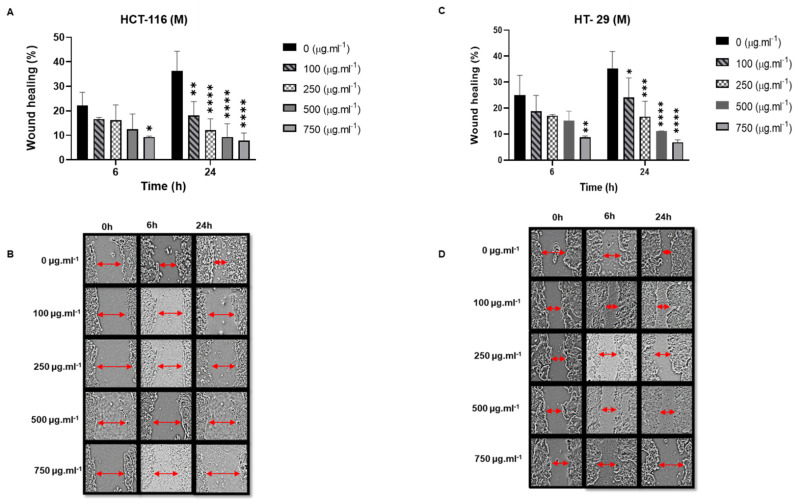
Wound-healing assay showing that M Soxhlet (100–250–500–750 µg.mL^−1^) inhibits the migration of HCT-116 and HT-29 cells after treatment for 24 h. (**A**,**B**) M Soxhlet treatment of HCT-116 cells. (**A**) Percentage of wound-healing at 6 h and 24 h. (**B**) Images of scratched HCT-116 cells taken at 0 h, 6 h, and 24 h. (**C**,**D**) M Soxhlet treatment of HT-29 cells. (**C**) Percentage of wound-healing at 6 h and 24 h. (**D**) Images of scratched HT-29 cells taken at 0 h, 6 h, and 24 h. Representative images were captured by a Leica microscope. Data are reported as mean ± SD, *n* = 3. * *p* < 0.05, ** *p* < 0.01, *** *p* < 0.001, and **** *p* < 0.0001 vs. control group. M: methanol.

**Table 1 molecules-27-06617-t001:** Proximal analysis of *J. rubens*.

Proximal Analysis	Results % (*w*/*w*)	SD
Ash Content	66.08	1.59
Humidity	2.31	0.46
Water-Soluble Ash	23.26	1.67
Acid-Insoluble Ash	24.04	2.21
Total Carbohydrates	14.48	6.63
Total Lipids	4.5	2.13
Total Proteins	11.3	1.74

Values are reported as mean ± SD; *n* = 3 refers to three independent experiments. SD: standard deviation.

**Table 2 molecules-27-06617-t002:** Elemental composition of *J. rubens* by atomic absorption spectroscopy.

Trace Minerals	Concentration (mg/g)
Fe	0.4 ± 0.04
Cu	0.03 ± 0.01
Zn	0.036 ± 0.13
Ca	24 ± 0.5
Mg	33 ± 0.5

Values are reported as mean ± SD (*n* = 3). Fe: iron; Cu: copper; Zn: zinc; Ca: calcium; Mg: magnesium.

**Table 3 molecules-27-06617-t003:** Quantitative analysis of the photosynthetic pigments present in *J. rubens*.

Type of Pigment	Concentration
Chlorophyll a (mg/g)	0.82 ± 0.02
Total Chlorophyll (mg/g)	1.04 ± 2.27
Chlorophyll c1 + c2 (µg/g)	4.68 ± 0.01
Carotenoids (mg/g)	0.14 ± 0.01

Values are reported as mean ± SD (*n* = 3).

**Table 4 molecules-27-06617-t004:** Antibacterial effect of *J. rubens* extracts.

Bacterial Strains	AQ Extract	DM Extract	M Extract	Positive Control
	Zone of Inhibition (mm)			
A. Baumannii (G-)	-	-	-	15
P. Aeruginosa (G-)	-	-	-	21
E. Coli (G-)	-	-	-	28
B. Cereus (G+)	-	-	-	28
S. Pneumonia (G+)	-	-	-	28

G+: Gram-positive; G-: Gram-negative. Positive control: penicillin/streptomycin. Data were reported as mean ± SD (*n* = 3). DM: dichloromethane/methanol; M: methanol; AQ: aqueous; SD: standard deviation.

**Table 5 molecules-27-06617-t005:** Cytotoxic activity of organic extracts on CRC cell lines.

	In Vitro Cytotoxicity, IC_50_ (µg/mL)	
Extracts	HCT-116	HT-29
DM Soxhlet	499.94	531.44
DM Crude	2603.54	1560.38
M Soxhlet	389.73	666.82
M Crude	1578.13	1662.74
AQ Soxhlet	1490.37	1657.26
AQ Crude	1629.45	1240.42

Data were reported as mean ± SD (*n* = 3). DM: dichloromethane/methanol; M: methanol; AQ: aqueous. SD: standard deviation. IC_50_ refers to the half-maximal inhibitory concentration.

## Data Availability

All data and materials support our published claims and comply with the field standards. The original data can be made available upon request.

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
