# Peer review of "The Lebanese Red Algae Jania rubens: Promising Biomolecules against Colon Cancer Cells"

_molecules, 2022, doi:10.3390/molecules27196617_

Round 1

Reviewer 1 Report

1. The English need improvement since there are some grammatical and syntax errors in the manuscript. For example,

·         in line number 24, the word “proximal” may be as “the proximal”;

·         in line number 87, “Rubens of” as “rubens on”;

·         in line number 167, “of at” as “of”;

·         in line number 171, “treatment the” as “treatment of the”;

·         in line number 176, “hours treatment” as “hours of treatment”;

·         in line number 223, “extracts is” as “extracts are”;

·         in line number 226, “HCT” as “the HCT”;

·         in line number 242, “treated” as “were treated”;

·         in line number 251, “Leica” as “a Leica”;

·         in line number 275, “a higher” as “higher”;

·         in line number 305, “antioxidant” as “an antioxidant”;

·         in line number 338, “New treat-ment” as “The new treatment”;

·         in line number 361, “grounded” as “ground”;

·         in line number 402, “then” as “and then”.

The grammar mistakes which are not mentioned here are also to be checked and corrected properly.

2. There are some typing mistakes as well, and authors are advised to carefully proof-read the text. For example,

·         in line number 44, the words “important-ly” may be as “importantly”;

·         in line number 45, “mul-tiple” as “multiple”;

·         in line number 56, “Medi-terranean .” as “Medi-terranean.”;

·         in line number 58, “Alex-andria” as “Alexandria”;

·         in line number 77, “Leba-nese” as “Lebanese”;

·         in line number 87, “anal-ysis” as “analysis”;

·         in line number 105, “Table2” as “Table 2”;

·         in line number 176, “dose dependent” as “dose-dependent”;

·         in line number 227, “that HT-29” as “that HT-29”;

·         in line number 275, “abundancy” as “abundance”;

·         in line number 281, “polysac-charides” as “polysaccharides”;

·         in line number 281, “seaweeds” as “seaweed”;

·         in line number 282, “infil-tration” as “infiltration”;

·         in line number 289, “regula-tion” as “regulation”;

·         in line number 291, “play” as “plays”;

·         in line number 292, “Calci-um” as “Calcium”;

·         in line number 294, “magnesi-um” as “magnesium”;

·         in line number 300, “con-tent” as “content”;

·         in line number 307, “can-cer” as “cancer”;

·         in line number 316, “extrac-tion” as “extraction”;

·         in line number 319, “me-tabolite” as “metabolite”;

·         in line number 321, “an-ti-migratory” as “anti-migratory”;

·         in line number 324, “antimigratory” as “anti-migratory”;

·         in line number 343, “in-cluding” as “including”;

·         in line number 345, “appa-ratus” as “apparatus”;

·         in line number 353, “Interest-ingly” as “Interestingly”;

·         in line number 367, “wa-ter” as “water”;

·         in line number 375, “meth-od” as “method”;

·         in line number 383, “precipi-tated” as “precipitated”;

·         in line number 395, “al-gae” as “algae”;

·         in line number 397, “ex-tracts” as “extracts”;

·         in line number 402, “so-dium” as “sodium”;

·         in line number 425, “ob-tained” as “obtained”;

·         in line number  426, “Pseu-domo” as “Pseudomo”;

·         in line number  454, “analy-sis” as “analy-sis”;

·         in line number  460, “quanti-tative” as “quantitative”.

The typos not mentioned here are also to be checked and corrected properly.

3. Check the abbreviations throughout the manuscript and introduce the abbreviation when the full word appears the first time in the text and then use only the abbreviation (For  example, Colorectal cancer (CRC), DPPH, MTT, etc.,). And it should be in both abstract as well as in the remaining part of the manuscript. Make a word abbreviated in the article that is repeated at least three times in the text, not all words need to be abbreviated.

4. The full form of the species should be given when the first time appears in both the abstract and in the remaining part of the manuscript and it should be followed by only the first letter of the genus (e.g., Jania rubens when the first time appears and followed by J. rubens). The genus an species name should be italic all over the manuscript and also the species name should be starts with small letter.

5. In the introduction, the authors should cite recent reference for the incidences of cancer since they cited 2019 and also the data is related with the lung cancer but not the colorectal cancer.

6. The figure legends should be improved and a proper footnote should be given. All legends should have enough description for a reader to understand the figure without having to refer back of the main text of the manuscript. For example, the necessary expansion may be given for abbreviations used.

7. In the materials and methods, the author should include the source of chemicals used in this study in separate heading.

8. The superscript and subscript have to be given properly (For example, superscript “IC50” “50” should be subscripted). And it should be thoroughly checked all over the manuscript.

9. The technical terms (Latin Phrase) “in vitro and in vivo (in reference, if applicable)” should be italic and it should be checked all over the manuscript.

10. The reference should be cited properly, for example, i) few reference full name of the journals given and for others only short form has been given ii) the first letters of journal name should be capital, iii) for few reference DOI number is given, but not for other references iv) and all the references to be corrected as per the journal format.

Author Response

Dear Editor,  

According to your instructions and the reviewers’ comments, we have revised the manuscript carefully, and we made the following corrections:  

Regarding the comments of reviewer 1:  

The grammatical and syntax errors were corrected.  
Typing mistakes were reviewed and corrected.  
The abbreviations were checked throughout the manuscript. The full name was added before the abbreviations appeared for the first time in the text.  
The name of the alga was corrected to Jania rubens at the beginning, followed by J. rubens all over the text.  
The citation nb. 16 was corrected. The new citation was published in 2020 and includes projection studies about colorectal cancer in Lebanon till 2025.  
The figures and table legends were reviewed and corrected with detailed explanations. In addition, full names were added to the abbreviations.  
The chemicals used and their source was added as a separate heading in the materials and methods.  
The superscripts and subscripts were checked all over the manuscript. (IC50 was corrected to IC50)  
The word in vitro was corrected to italic.  
The references were thoroughly reviewed and corrected according to the journal format.  

Reviewer 2 Report

 The paper is correctly written.

I have few minor and one or two major objections.

Lines 53, 58 - Medi-terranean, Alex-andria ® Mediterranean, Alexandria. There are much more such broken words in the text.

Line 97 – Define N in Table 1

Lines 271 – 286 – Consider re-writing the whole paragraph because it is in the present form more suitable for Introduction.

 My main objective is the lack of specificity. More precisely, the cell viability assay (MTT-based method) should have been also applied on a healthy cell line and/or on another cancer cell line.

In addition, the active substance(s) remain(s) undiscovered, but that can be the subject of further studies.

Author Response

Dear Editor,  

According to your instructions and the reviewers’ comments, we have revised the manuscript carefully, and we made the following corrections:  

Regarding the comments of reviewer 2:  

The broken words and typing errors were corrected.  
N=3 in table one was defined as three independent experiments  
The paragraph from line 271 to line 286 was reviewed and rewritten to address the idea more precisely.  
Our current study is based on testing the effective cytotoxic extracts of Jania rubens on other cell lines like MCF7 and MDA breast cancer cells. We are also working on identifying the active biomolecules through different isolation techniques.   

Round 2

Reviewer 1 Report

1. There are some grammatical, alignment and typographical errors are noted in the manuscript and it should be thoroughly checked and corrected throughout the manuscript. For example, the words “an insight” may be as “insight”; “Medi-terranean .” as “Mediterranean.”; “Izmir” as “the Izmir”; “Tunisian” as “the Tunisian”; “Cu: Cupper” as “Cu: Copper”; “as positive” as “as a positive”; “Gram positive” as “Gram-positive”; “Gram negative” as “Gram-negative”; “crude induce” as “crude induces”;  “Data was reported” as “Data were reported”; “IC50 equals” as “IC50 equal”; “time dependent” as “time-dependent”; “and soxhlet” as “and the soxhlet”; “increase the” as “increases the”; “the Lebanese” as “Lebanese”; “and carbohydrates” as “and carbohydrate”; “possess effective” as “possess an effective”; “lipid, carbohydrate, and protein” as “lipids, carbohydrates, and proteins”; “protects from” as “protects against”; “extracts is” as “extracts are”;  “by Soxhlet” as “by the Soxhlet”; “by concentrated” as “with concentrated”; “dichloromethane / methanol” as “dichloromethane/methanol”; “temperature” as “temperatures”; “penicillin streptomycin” as “penicillin-streptomycin”; “Acknowledgments” as “Acknowledgements”.

2. The authors should do the following in both abstract and the remaining part of the manuscript. Check the abbreviations throughout the manuscript and introduce the abbreviation when the full word appears the first time in the text and then use only the abbreviation. For  example, Colorectal cancer (CRC) is given both expansion and abbreviation when first time appear in the abstract and remaining part and in line numbers, 525 and 552. This has to be checked all other abbreviations used in the manuscript. 

Author Response

Comments and Suggestions for Authors

  1. There are some grammatical, alignment and typographical errors are noted in the manuscript and it should be thoroughly checked and corrected throughout the manuscript. For example, the words “an insight” may be as “insight”; “Medi-terranean .” as “Mediterranean.”; “Izmir” as “the Izmir”; “Tunisian” as “the Tunisian”; “Cu: Cupper” as “Cu: Copper”; “as positive” as “as a positive”; “Gram positive” as “Gram-positive”; “Gram negative” as “Gram-negative”; “crude induce” as “crude induces”;  “Data was reported” as “Data were reported”; “IC50 equals” as “IC50 equal”; “time dependent” as “time-dependent”; “and soxhlet” as “and the soxhlet”; “increase the” as “increases the”; “the Lebanese” as “Lebanese”; “and carbohydrates” as “and carbohydrate”; “possess effective” as “possess an effective”; “lipid, carbohydrate, and protein” as “lipids, carbohydrates, and proteins”; “protects from” as “protects against”; “extracts is” as “extracts are”;  “by Soxhlet” as “by the Soxhlet”; “by concentrated” as “with concentrated”; “dichloromethane / methanol” as “dichloromethane/methanol”; “temperature” as “temperatures”; “penicillin streptomycin” as “penicillin-streptomycin”; “Acknowledgments” as “Acknowledgements”.

  1. The authors should do the following in both abstract and the remaining part of the manuscript. Check the abbreviations throughout the manuscript and introduce the abbreviation when the full word appears the first time in the text and then use only the abbreviation. For  example, Colorectal cancer (CRC)is given both expansion and abbreviation when first time appear in the abstract and remaining part and in line numbers, 525 and 552. This has to be checked all other abbreviations used in the manuscript. 

All reported comments have been taken into consideration and corrected in the manuscript.

Reviewer 2 Report

N is now defined. It is not necessary to write N= in the corresponding table columns!

Author Response

Comments and Suggestions for Authors

N is now defined. It is not necessary to write N= in the corresponding table columns!

Done
